# Effect of Different Timings of Implant Insertion on the Bone Remodeling Volume around Patients’ Maxillary Single Implants: A 2–3 Years Follow-Up

**DOI:** 10.3390/ijerph17186790

**Published:** 2020-09-17

**Authors:** Giovanni Battista Menchini-Fabris, Paolo Toti, Giovanni Crespi, Ugo Covani, Luca Furlotti, Roberto Crespi

**Affiliations:** 1Department of Multidisciplinary Regenerative Research, Guglielmo Marconi University, Via Vittoria Colonna, 11, 00193 Rome, Italy; gb.menchinifabris@gmail.com (G.B.M.-F.); robcresp@libero.it (R.C.); 2San Rossore Dental Unit, Viale delle Cascine 152 San Rossore, 56122 Pisa, Italy; 3Department of Stomatology, Tuscan Stomatological Institute, Foundation for Dental Clinic, Research and Continuing Education, Via Padre Ignazio da Carrara 39, 55042 Forte Dei Marmi, Italy; gio.crespi@hotmail.it (G.C.); covani@covani.it (U.C.); luca.furlotti94@gmail.com (L.F.)

**Keywords:** cone beam computerized tomography imaging, immediate placement, delayed loading

## Abstract

*Background*: To investigate the middle-term effect on bone remodeling of different timings for different implant placement (immediate versus delayed). *Methods*: Patients with an anterior maxillary failing tooth were treated by single-crown supported by dental implant. Subjects were retrospectively analyzed for 3 years and assigned to one of two predictor groups: nine immediate versus 10 delayed implant placement (1–2 months after tooth extraction). The crestal bone loss around dental implants was measured with the cone beam computerized tomography by fusing pre-operative and post-operative data. *Results:* The percentage of volume loss registered at 1-year follow-up (%ΔV) was of 7.5% for the immediate group, which was significantly lower (*p*-values ≤ 0.0002) than the loss of 24.2% for the delayed group. At 3 years, there was a significant difference (*p*-values = 0.0291) between the two groups, respectively, with a volume loss of 14.6% and 27.1%. When different times were compared, the percentage of the volume loss for the immediate group was different (*p*-value = 0.0366) between the first and third year (7.5% and 14.6%, respectively). For the delayed group, no significant difference was registered between the 1- and 3-year follow-up. *Conclusions*: The bone loss around dental implant-supported single-crown with different timing of insertion appeared higher for the delayed group than the immediate group.

## 1. Introduction

The ultimate goal of the clinician is to counteract the effect of tooth extraction on bone which begins the slow process of remodeling just after avulsion. With regard to fixed prosthetic rehabilitation using dental implant, several years have passed since authors suggested that an immediate loading of implant positioned into fresh socket avoids collapse of the alveolar tissue, and guarantees an optimal aesthetic outcome, in particular for the soft tissue profile [1,2]. Reviews of the timing of dental implant placement suggest that immediate- and delayed-implant placement exhibit similar behavior when the survival rate and tissue remodeling are similar have been compared between the two groups [3,4]. It has been suggested that tooth extraction procedure (in the least disruptive way possible) has a major effect on clinical outcomes, whereas the dental implant positioning may have just a minimal effect [5].

The criteria for applying either immediate or conventional dental implant surgery are different. The delayed implant placement requires complete healing of the extraction socket and second surgery consisting of flap-approach, whereas a dental implant can be immediately placed via a flapless surgery and a preservation of the bony walls of the socket. As stated, a minimization of the bone loss for the prevention of soft tissue recession can be obtained by avoiding the fracture of the buccal plate even for conventional dental implant placement [6].

Published studies show that timing of implant placement after tooth extraction can affect the peri-implant linear bone loss; however, information that defines the event in all its dimensions, with particular attention to volumetric bone change associated with the timing of implant placement, is scant [7].

Cone beam computerized tomography (CBCT) is a very suitable diagnostic instrument for the planning of prosthetic rehabilitation supported by implants; moreover, the comparison of the pre-operative and post-operative scans can be used for the analysis of three-dimensional changes of the peri-implant tissues [8].

The primary aim of this paper is to investigate how different timings of implant placement (immediate versus delayed) affect the bone volume of single dental implants placed in maxilla within a 3-year follow-up. The secondary aim is to test the effect on bone loss of tooth positions (incisor or premolar).

## 2. Materials and Methods

### 2.1. Study Population and Design

In this retrospective comparative study, the considered patients’ population was treated from 2010 to 2016. Data were acquired by reviewing each patient’s case sheet. Informed consent was signed by all individual participants included in this data analysis. The preparation of the manuscript followed the STROBE statement. All procedures performed in studies involving human participants are in accordance with the ethical standards of the institutional and/or national research committee and with the 1964 Helsinki declaration and its later amendments or comparable ethical standards. For this type of study formal consent is not required.

#### 2.1.1. Inclusion Criteria

Adult patients (greater than or equal to 18 years);Patients underwent single-tooth extraction of the incisor or premolar tooth due to caries, endodontic failure, periodontal disease without reported defect of bony wall;Patients underwent immediate or delayed dental implant placement with delayed prosthetic rehabilitation;Patients with a full set of preoperative (before tooth extraction) and postoperative 3D imaging.

#### 2.1.2. Exclusion Criteria

Metal corrupted CBCT scans;Patients who had received irradiation and/or bone resection as part of a cancer treatment plan;Patients who received bisphosphonates (intravenous and/or oral).

The cone-beam computerized tomography scan (Gendex GXCB-500; Gendex Dental Systems 1910 North Penn Road Hatfield, PA, USA) provide finely-detailed three-dimensional overview of the bony architecture and the associated anatomical structures. The following set of default parameters have been applied in the acquisition of scans: 120 kV, 30.89 mAs, 0.2 mm × 0.2 mm × 0.2 mm isotropic voxel size, and 8.72 mm Ø of FOV.

As part of the standard treatment guidelines, the surgery was routinely planned via virtual three-dimensional diagnostic tool using a preoperative CBCT scan (baseline or before tooth extraction). Before any surgery following the implant placement, the clinician needs to prescribe an adjunctive CBCT (postoperative).

A routine showed in Appendix A allows a Matrix Laboratory (Image Processing Toolbox, MatLab 7.1; The MathWorks, Natick, MA, USA) to read and save the CBCT scan data and to fuse preoperative and postoperative axials. First of all, position and angulation of preoperative and postoperative data are changed in space, in order to better evaluate the volume remodeling as per accessory and core algorithms listed in Appendix A. An intrapatient superimposition of two CBCT scans allows two voxels (usually preoperative and postoperative) to occupy the same space at the same time. The accuracy of superimposition has been measured as per Crespi and co-workers [9] via three reference points: the apex of the anterior nasal spine (ANS), and the two hamulus pterygoideus (HP right and HP left). Then fused files have been saved in a single .dicom file (Digital Imaging and Communications in Medicine) (Figure 1) [10]. At the end of elaboration, volumes (V) are measured as per Sbordone and co-workers [11] within a standardized volume of interest (VOI) contained within the following boundaries: mesial tooth, distal tooth, and 10 mm apically to the most coronal portion of bone level measured at baseline. All measurements have been single blinded clinician, not involved in surgery, performed all the volume measurements.

### 2.2. Surgical Procedure

#### 2.2.1. Immediate Placement

One hour prior to surgery, patient receives 1 g amoxicillin and then 1 g twice a day for a week after the surgical procedure. Surgery is performed under local anesthesia with optocaine 20 mg/mL with adrenaline 1:80.000. The tooth is extracted with a magnetic device (Magnetic Mallet, www.osseotouch.com, Turbigo, Milano, Italy), which allows the clinician to maintain the integrity of the socket with a flapless approach [12]. Immediate implant placement requires an absolute lack of fenestration and dehiscence in the socket walls. The implant site is prepared with standard drills following the palatal bony walls as a guide, as per instruction of the manufacturers. The apical portion of the implant is always placed engaging the pristine bone of the socket beyond the root apex. No countersinking was used. External-hexagon osseointegrated dental implant with rough titanium plasma spray surface, 0.8 mm machined neck and progressive thread design (out-Link PRO-Link: Sweden & Martina, Due Carrare, PD Italia), is positioned without filling voids between implant surfaces and alveolus walls. The mucosal free margins are firmly sutured to the periosteum, in order to stop the bleeding and to ensure blood clot stability. A secondary intention healing was pursued.

#### 2.2.2. Delayed Placement

The implant is placed 1–2 months after tooth extraction. Patients receive a prophylaxis antibiotic, 1 g of amoxicillin and clavulanic acid (or 600 mg clindamycin if allergic to penicillin) 1 h prior to implant placement. Patients receive local anesthesia using optocaine 20 mg/mL with adrenaline 1:80.000. A partial thickness flap is raised, and the implant is placed. The implant osteotomy sites are prepared according to the manufacturer’s recommendations. The external-hexagon osseointegrated dental implant with rough titanium plasma spray surface, 0.8 mm machined neck and progressive thread design (out-Link PRO-Link: Sweden & Martina, Due Carrare, PD Italia), is positioned. The cover screw is placed and flaps are sutured with resorbable 4-0 suture. All patients are prescribed ibuprofen 600 mg 3 times per day for as long as required, and chlorhexidine mouthwash 0.2% twice a day for 15 days.

### 2.3. Prosthetic Procedure

After 3 months of submerged healing, implant is exposed to the oral cavity and impressions are taken with the transfer/abutment using a tray with polyvinyl siloxane material. A healing screw is positioned to promote the maturation of soft tissues and, 4 months after healing, a precision impression is taken with a polyvinyl siloxane material (Flexitime, Heraeus/Kulzer, Hanu, Germany). A final metal-ceramic crown is cemented on personally tailored titanium abutment. Patients received yearly checkups to monitor the state of health of their dental elements. Clinicians recommended to patients a professional oral hygiene session every six months.

### 2.4. Variables

Variables are divided into anatomical variables, primary predictor variable, and outcome variables. Variables for sample description—age, gender, and tooth position—are registered for each patient.

#### 2.4.1. Anatomical Variables

Anatomical measurements related to the volume of the treated sites are acquired and processed, to obtain all of the outcome variables. The anatomical variables are positive in value.

V: volume of the alveolar ridge is obtained through the standardized VOI at baseline (V_preop_) and at post-operative survey (V_postop_).

#### 2.4.2. Primary Predictor Variable

Immediate group: immediate dental implant positioning and delayed non-functional loading (immediate); or delayed group: delayed dental implant positioning and delayed loading (delayed).

#### 2.4.3. Secondary Predictor Variable

Tooth position, which is incisor or premolar site (bicuspid).

#### 2.4.4. Outcome Variables

The outcome variables for alveolar ridge modification can be either positive or negative. A negative number indicates that the variable describes a reduction in the bone volume:

ΔV: alveolar ridge bone volume changes from baseline to either the 1-year time point (preop→1 yr) or the 3-year time point (preop→3 year), evaluated by subtracting the respective baseline value (preop) from the post-operative measurement (postop).
(1)$$\Delta V_{\mathrm{postop}}=V_{\mathrm{postop}}-V_{\mathrm{preop}}$$

%ΔV: Percentage of the alveolar ridge bone volume changes, evaluated at 1 and at 3 years (V_postop_), is the ratio between the ΔV and the volume of the alveolar ridge at baseline (V_preop_), following Equation (2).
(2)$$\%\Delta V_{\mathrm{postop}}=100\times \frac{V_{\mathrm{postop}}-V_{\mathrm{preop}}}{V_{\mathrm{preop}}}$$

SR: an implant failure is defined as removal of the dental implant due to pain or implant mobility caused by peri-implant bone loss and/or infection. The presence of implant mobility and/or pain is evaluated by applying force to the single-crown with two metallic handles of dental instruments. Survivals are calculated according to Romeo and colleagues [13].

### 2.5. Statistical Analysis

All patient-related data were entered into a database (Database Toolbox, MatLab 7.1; The MathWorks, Natick, Massachusetts, USA), allowing calculations to be performed automatically. Statistical analyses were performed using a statistical tools package (Statistics Toolbox, MatLab 7.11; The MathWorks). The two groups were independent (one site per patient was selected), but the Shapiro-Wilk test failed to confirm the normal distribution of all sub-variables in both groups (Table 1). Variable pair-wise comparisons were performed by the Wilcoxon rank-sum test for unmatched data (between groups, obtaining procedures related *p*-value, p_W_).

Measurements in the tables are described as median and interquartile range. All results were rounded to the nearest decimal. The level of statistical significance was set at 0.05.

## 3. Results

Retrospective present information about forty-seven patients (one implant per patient) were initially gathered from the CBCT scan archives; five patients were excluded from the analysis, because their CT images appeared corrupted by the metal artifacts. Another one patient was excluded for the presence of a residual palatal root after dental implant positioning, as seen on radiograph. Forty-one dental implants were finally included. At 1-year survey, nine patients were subjected to tooth extraction with immediate dental implant insertion, immediate, whereas 10 patients underwent delayed dental implant positioning, delayed (Table 1 and Figure 2). Three-year demographic and volumetric survey of 22 patients included (11 in the immediate group and 11 in the delayed group) are shown in Table 2 and Figure 3.

No postoperative complications were registered in any of the selected patients. At the time of the first surgery, 41 patients were examined (mean age of 46.1 ± 8.4 years, range from 28.7 to 66.2 years; 18–30 years: one patient; 30–40 years: eight patients; 40–50 years: twenty-one patients; 50–60 years: nine patients; 60–70 years: two patients); 16 were male and the percentage of incisors and predator were, respectively, 60% and 40% in the immediate group, and 57.1% and 42.9% in the delayed group (Table 1 and Table 2).

The accuracy of the superimposition of three-dimensional tomographic scans was less than 1 mm.

### 3.1. Loading Procedure

Table 1 and Table 2 reported anatomical (V_preop_, V_1 yr_ and V_3 yr_) and outcome variables (ΔV and %ΔV); moreover, significances between the different time points of implant placement (immediate versus delayed) were also reported.

At 1 year, the two procedures were compared: the volume loss registered in the immediate placement group (0.04(0.02) cc with a percentage of bone loss of 7.5(3.1) %) was significantly lower (*p*-values ≤ 0.0002) than that registered in the delayed group (0.15(0.07) cc with a %ΔV of −24.2(12.0) %).

At the 3-year point of the survey, a significant difference (*p*-values = 0.0350) was registered between the two groups regarding the volume loss (preop at 3 year), which was 0.07(0.11) cc for the immediate group and 0.18(0.10) cc for the delayed placement; moreover, the percentage of volume loss at 3-year follow-up in the immediate group [14.6(15.5)%] was significantly lower (*p*-value = 0.0291) than that registered for the delayed group [27.1(11.7)%].

### 3.2. Time of Implant Placement

When different times of implant placement had been compared, for the immediate group significant differences were registered between the volume at 1-year ($V_{1\;\mathrm{yr}}$ of 0.54(0.07) cc) and the volume at 3-year ($V_{3\;\mathrm{yr}}$ of 0.46(0.09) cc), with a *p*-value of 0.0119, and between the percentage of the volume loss at 1-year survey (%ΔV preop at 1 year of 7.5(3.1)%) and that measured at 3-year point of the survey (%ΔV preop at 3 year of 14.6(15.5)%), with a *p*-value of 0.0366. No significant difference was seen when the times of the delayed group had been compared.

### 3.3. Implant position

When dental implant positions (incisor versus bicuspid) were compared, the subgroups showed very similar behaviors at the 1- and 3-year points of the survey (Table 1 and Table 2, and Figure 2 and Figure 3), with no significant differences. However, at the 3-year point of the survey, for the immediate group the percentages of volume loss were 10.9(12.1)% in the incisors and 24.5(3.8)% in the bicuspids.

## 4. Discussion

The purpose of this retrospective study was to evaluate and compare immediate versus delayed implant placement in terms of the volume loss of bone around single dental implants rehabilitated with delayed fixed prostheses.

The literature review has shown that only a very limited number of randomized clinical trials compared immediate and delayed implants; moreover, no controlled studies are available on the role of timing of implant placement after tooth extraction to evaluate the effect on volume loss of bone around single dental implants in patients rehabilitated with delayed fixed prostheses. In this study, no implant failure was encountered in 3 years of survey. This data confirmed results of the other authors, in which middle-term survival rates for immediate placement/conventional loading of dental implants ranged between 95% and 100%; whereas for conventional placement/conventional loading of dental implants, survival rates ranged from 93% to 100%) [14,15]. The tendency was also confirmed by a review in which results of the weighted meta-analysis indicated no significant advantage in survival rates at 1-year survey of implants conventionally restored neither in favor of the immediate/early-placed implants nor the implants conventionally placed [16].

Although the preservation of soft and hard tissue is the rationale for immediate implant placement in a fresh extraction socket, the extent of preservation is, nevertheless, nowadays a subject of discussion, as reported in some papers [17].

As said, in this study the immediately placed implants showed lower bone volume bone loss (less than 10%) compared with the loss registered for implants placed with conventional insertion technique (in healed bone with delayed loading), with a loss of 24% of the initial volume of interest. Volume loss measured at 3-year survey showed that, again, the percentage of bone loss in the delayed group (27.1%) is something over two times the percentage of bone loss in the immediate group (14.6%); one significant difference remains.

This confirmed that immediate placement and restoration of a single implant was a successful option of treatment in the case of single compromised teeth, as attested by previous authors with good results in terms of short/long term success (close to 100%) and marginal bone loss (from 0.42 to 2.69 mm) [18,19]. Moreover, immediate placement of dental implant protocol seems to maintain the preexisting architecture of soft and hard tissues in most cases, as reported in the literature [20].

The effectiveness of the replacement of missing teeth with immediate implant placement in terms of aesthetic and functional success has now been clearly established by some review studies, in spite of a high degree of heterogeneity in measurement methods of bone level [21,22,23]; all clinicians could take a different baseline for the bone loss measurements in their radiographic examinations.

As described, damage to the buccal bone plate is one of the easiest factors to control with a view to obtaining alveolar ridge preservation; atraumatic extraction is mandatory, to the point where multi-rooted tooth may be sectioned before extraction [17].

Even if randomized trials describing the result from the bone volume analysis are not available for some treatment comparisons, that is, in particular between immediate versus delayed placement of implants, the results concerning linear remodeling have echoed the trend already seen in the present paper regarding bone loss in the esthetic zone. In fact, in the study by Pellicer-Chover [24], twelve months after loading, the linear bone loss for immediate and delayed placement of implants were 0.54 ± 0.39 mm and 0.66 ± 0.25 mm, respectively, without any significant difference between the two groups.

Moreover, none of the studies describing the linear remodeling within the first three years after implant loading [14], revealed a significant difference for the bone loss between the immediate implant placement group (with a loss ranging from 0.35 to 1.5 mm) and the delayed implant placement group (with a loss ranging from 0.42 to 1.4 mm).

The bone volume change trend described in the present paper was in part confirmed by the study of Tonetti and co-workers, in which, after loading, immediately placed implants showed a significantly more pronounced bone remodeling, compared to delayed implants placed with conventional insertion technique [25]. After the first year of loading, the remodeling in the delayed implant placement group remained stable over time, whereas the bone loss in the immediate implant placement group continued its slight decrease over time, up to three years, according to the survey; however, the linear remodeling did not show statistical differences between groups [25].

The present retrospective study was primarily limited by the lack of a treatment selection. The type of dental implant placement was neither masked nor randomized. Ranking of patients in the two groups (immediate versus delayed placement of implant) was done on the basis of the information on the medical chart. Given the nature of the study, it should be noted that the presence of data regarding bone volume remodeling remains one of the most important criteria. Moreover, externalization of results to cohorts with a different tooth position is not possible.

On the other hand, the strength of the present study was the single brand and type of dental implant and the uniformity of surgical performances. Any final confirmation of the present findings will require longer periods of observation and higher number of enrolled implants.

## 5. Conclusions

The present study attested that the bone volume loss of the crestal bone around a dental implant-supported single-crown with different timings of insertion appeared higher for the delayed group than the simultaneous group. The immediate implant placement group showed more predictable results after restoration, even if it failed to arrest bone loss around dental implant.

## Figures and Tables

**Figure 1 ijerph-17-06790-f001:**
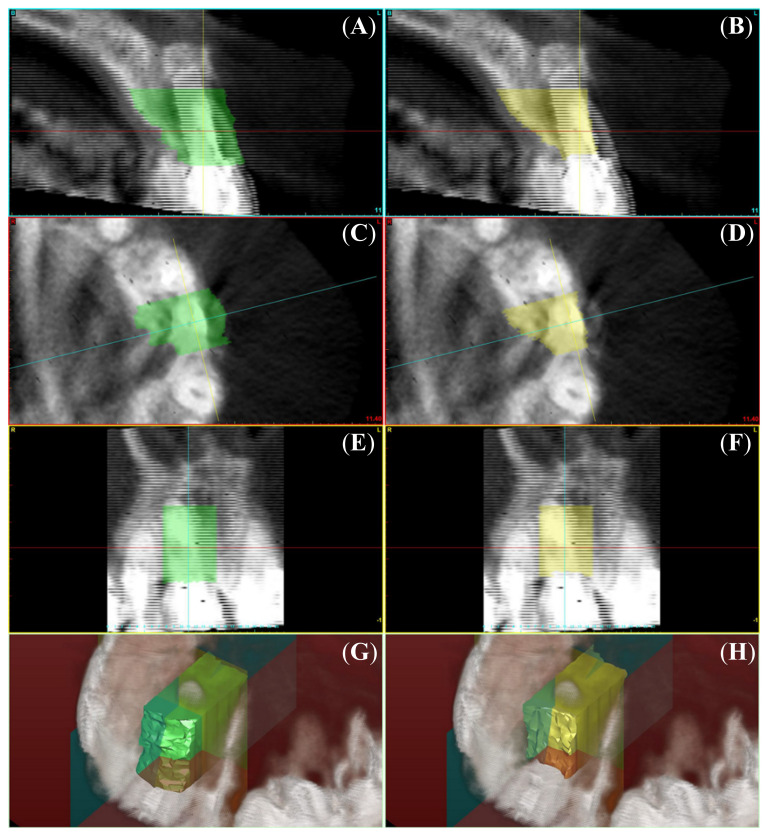
Superimposed views of both pre-operative and post-operative scans: measured pre-operative Volume of Interests (VOIs) on the left (**A**) in cross-sectional view; (**C**) in axial view; (**E**) in panorex view; (**G**) in volume rendering, and measured post-operative VOIs on the right (**B**) in cross-sectional view; (**D**) in axial view; (**F**) in panorex view; (**H**) in volume rendering. Minor ticks at the bottom of each image are millimeters.

**Figure 2 ijerph-17-06790-f002:**
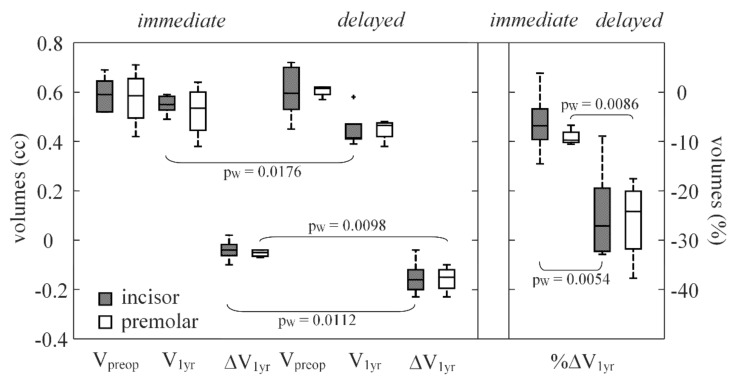
Box plots for the anatomical and outcome variables described for baseline (preop) and 1-year postoperative time (1 year) and procedure applied (immediate or delayed); volume of alveolar ridge or V into the VOI, and outcome variable (alveolar ridge volume change or ΔV_1 yr_, and its percentages or %ΔV_1 yr_). In box-and-whiskers plot, the box line represents the lower, median, and upper quartile values, the whisker lines include the rest of the data. Outliers were data with values beyond the ends of the whiskers. Wilcoxon rank-sum test significance between unpaired data (p_W_).

**Figure 3 ijerph-17-06790-f003:**
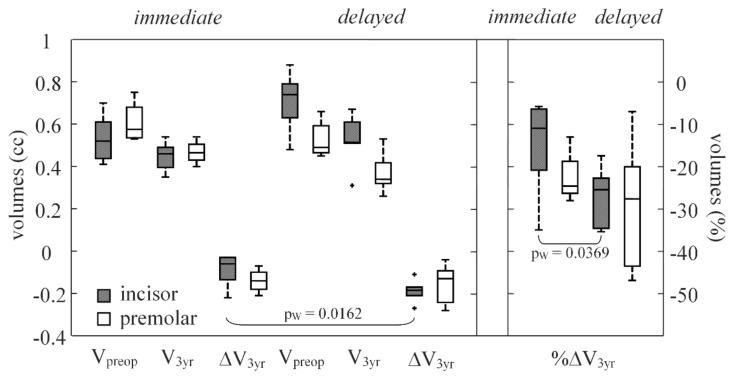
Box plots for the anatomical and outcome variables described for baseline (preop) and 3-year postoperative time (3 yr) and procedure applied (immediate or delayed); volume of alveolar ridge or V into the VOI, and outcome variable (alveolar ridge volume change or ΔV_3 yr_, and its percentages or %ΔV_3 yr_). In box-and-whiskers plot, the box line represents the lower, median, and upper quartile values, the whisker lines include the rest of the data. Outliers were data with values beyond the ends of the whiskers. Wilcoxon rank-sum test significance between unpaired data (p_W_).

**Table 1 ijerph-17-06790-t001:** Demographic data for the two groups at 1 year of survey. Anatomical and outcome variables before and 1 year after surgery: volume of alveolar ridge or V into the VOI, and outcome variable (alveolar ridge volume change percentages or ΔV%). Shapiro-Wilk test significance (p_SW_), Wilcoxon rank-sum test significance between unpaired data (p_W_). Statistically significant values are in bold.

Procedure		Immediate			Delayed		
sample size		9			10		
genders ratio (male/female)		3/6			4/6		
*incisor/bicuspid*		5/4			6/4		
**Procedure**		**Immediate**			**Delayed**		
**Variable**	**Size**		**p_sw_**	**Size**		**p_sw_**	**p_w_** **Immediate vs. Delayed**
*Overall*							
$V_{\mathrm{preop}}$ (mm^3^)	9	0.59(0.11)	0.8799	10	0.61(0.05)	0.6422	0.6526
$V_{1\;\mathrm{yr}}$(mm^3^)	9	0.54(0.07)	0.2777	10	0.44(0.06)	**0.0994**	**0.0084**
ΔV (mm^3^) preop→1 year	9	−0.04(0.02)	0.2282	10	−0.15(0.07)	0.7607	**0.0002**
%ΔVpreop→1 year	9	−7.5(3.1)	**0.0465**	10	−24.2(12.0)	0.8311	**<0.0001**

**Table 2 ijerph-17-06790-t002:** Demographic data for the two groups at the third year of the survey. Anatomical and outcome variables before and 3 years after surgery: volume of alveolar ridge or V into the VOI, and outcome variable (alveolar ridge volume change percentages or ΔV%). Shapiro-Wilk test significance (p_SW_), Wilcoxon rank-sum test significance between unpaired data (p_W_). Statistically significant values are in bold.

Procedure		Immediate			Delayed		
sample size		11			11		
genders ratio (male/female)		4/7			5/6		
*incisor/bicuspid*		7/4			6/5		
**Procedure**		**Immediate**			**Delayed**		
**Variable**	**Size**		**p_sw_**	**Size**		**p_sw_**	**p_W_** **Immediate vs. Delayed**
**Overall**							
$V_{\mathrm{preop}}$ (mm^3^)	11	0.54(0.12)	0.7472	11	0.63(0.26)	0.3601	0.2297
$V_{3\;\mathrm{yr}}$ (mm^3^)	11	0.46(0.09)	0.6381	11	0.51(0.19)	0.4537	0.9676
ΔV (mm^3^) preop→3 year	11	−0.07(0.11)	**0.0452**	11	−0.18(0.10)	0.8955	**0.035**
%ΔVpreop→3 year	11	−14.6(15.5)	0.3206	11	−27.1(11.7)	0.9334	**0.0291**

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
