# Peer review of "Effect of Different Timings of Implant Insertion on the Bone Remodeling Volume around Patients’ Maxillary Single Implants: A 2–3 Years Follow-Up"

_ijerph, 2020, doi:10.3390/ijerph17186790_

Round 1

Reviewer 1 Report

" Effect of different timings of implant insertion on the bone remodeling volume around patients’ maxillary single implants: a 2-3 years follow-up” is a study that examined the effect on bone mass of a single dental implant placed in the maxilla within the mid-term follow-up of different timings of implant placement (immediate versus delayed). This clinical observational study is very interesting. However, there are corrections that are essential to meet the standard for publication. Please refer to the following comments.

  1. Patient age is an important factor for implants. However, the distribution of age is not clarified in this paper. Please indicate.

  1. This study does not provide selection criteria for immediate and post-extraction placement after tooth extraction. It is an important factor related to bias. Please show me.

  1. The standard follow-up for post-prosthesis maintenance is unknown.

Please add.

  1. This study has too few cases and very poor generalization.

Among the groups of tooth parts, there are variations among each group such as anterior teeth (central incisors, lateral incisors, canines) and premolars (first premolars, second premolars).

There is also no consideration of altered parts of the bone.

It's a great clinical paper to look at, but the study design and case size limits are too big.

Please reconsider your study design.

Author Response

First of all, I would like to express my appreciation for the comments and suggestions.

The manuscript has been revised according to the reviewer’s requests.

" Effect of different timings of implant insertion on the bone remodeling volume around patients’ maxillary single implants: a 2-3 years follow-up” is a study that examined the effect on bone mass of a single dental implant placed in the maxilla within the mid-term follow-up of different timings of implant placement (immediate versus delayed). This clinical observational study is very interesting. However, there are corrections that are essential to meet the standard for publication. Please refer to the following comments.

  1. Patient age is an important factor for implants. However, the distribution of age is not clarified in this paper. Please indicate.

Our response: According to reviewer's suggestion, the age range of the participants has been inserted as requested.

  1. This study does not provide selection criteria for immediate and post-extraction placement after tooth extraction. It is an important factor related to bias. Please show me.

Our response: This is a retrospective analysis. So no random enrollment of the patients could be performed. The patient’s data selection for this retrospective statistical study depended on the timing of dental implant placement as it resulted from patients’ case-sheets (dichotomic choice).  

  1. The standard follow-up for post-prosthesis maintenance is unknown. Please add.

Our response: The prosthesis maintenance program was a dental hygiene every six months; this is single implant prosthesis.

  1. This study has too few cases and very poor generalization.

Our response: The study involved 47 patients, numbers comparable to several clinical study in literature. Moreover, for each patient computerized tomography scans had been collected from his/her case-sheet. Three-dimensional investigation is not a routine examination for a follow-up.

Among the groups of tooth parts, there are variations among each group such as anterior teeth (central incisors, lateral incisors, canines) and premolars (first premolars, second premolars).

Our response: The immediate and delayed groups showed no significant different distribution regarding the gender and the tooth site, nevertheless there is presence of a valuable variation at 3-year survey in the apparent percentages of bone loss [10.9(12.1) % in the incisors and 24.5(3.8) % in the bicuspids]. A linear regression analysis with a number of subjects less than 100 would result in an ineffective statistical test.

There is also no consideration of altered parts of the bone.

Our response: The volume measurements were performed as objectively as possible. No judgment and experience of the practitioners can be extrapolated by the case-sheets except failure and clinical outcomes.

It's a great clinical paper to look at, but the study design and case size limits are too big. Please reconsider your study design.

Our response: Clinicians of the present paper adhered to standard treatment guidelines according to which it was possible to routinely prescribe CBCT for diagnosed and surgical planning (preoperative or before tooth extractions). An additional CBCT scan was required to determine the appropriate surgical approach in the event of additional clinical needs (postoperative). The limited number of data and analyses necessarily retrospective are part of the reason that design and size of the study appeared to have a serious limitation.

Reviewer 2 Report

Review on the article Effect of different timings of implant insertion on the bone remodeling volume around patients’ maxillary single implants: a 2-3 years follow-up, submitted to International Journal of Environmental Research and Public Health.

The article is well-written and should be considered to be published within the journal. However, during reading, some doubts about certain statements might appear. If possible, the authors should consider making slight changes within the manuscript or provide answers in a response letter. Nevertheless, these doubts do not reduce the overall, positive impression of the article.

  1. Abstract

I would suggest describing healing time period in the case of delayed implant placement.

  1. Materials and Methods

Font type in the last sentence of 2.1 section is different in comparison to the rest of the manuscript.

Was there any age of the patient that was excluding patient from the research? As presented in some studies, age can negatively influence healing possibilities and overall bone quality.

Was Volume of Interest subjected to any smoothing procedures? Initially created anatomical models are characterised most of the time by the presence of several discontinuities as well as unexpected curvatures.

According to “Implant is placed 1-2 months after tooth extraction” – a month of difference in delayed placement method seems to be significant. Again, some studies present that bone density can change in a duration of a month by a few percent. Did authors observe differences in bone quality between patients subjected to surgeries after 1 and  2 month respectively?

In the last sentence of anatomical variables section, again there is a different font type in comparison to the manuscript style.

There is no real description of the implants used in the research. Can authors describe their types (press-fit/threaded) as well as manufacturing procedures with special focus on surface condition? Appropriate coating, surface roughness, material etc. should be described as these factors directly affect the possibility of achieving proper osseointegration.

  1. Results

The authors present patients mean age of 46.1 years. What was the age of the youngest and the oldest participant? Their age would significantly affect a mean result. I would suggest presenting, despite of mean age, the number of participants in intervals (i.e. 18-30 years: 3 patients, 30-40 years: 5 patients, etc.). This would give appropriate insight into the age of participants.

  1. Discussion

Successful osseointegration does not depend only on appropriate amount of bone around the implants. There are more factors that affect this phenomena, especially associated with implant construction. For this reason, I would suggest omitting information presented in the fourth paragraph.

Author Response

Dear Sirs

First of all, I would like to express my appreciation for the comments and suggestions offered by Editor and 4 Reviewers.

We have tried to address all the concerns expressed. To assist you in your work, we have highlighted all changes to the manuscript using the MS Word tracking function, though the modifications are so extensive as to make following the highlighted document all but impossible.

The manuscript has been revised according to the reviewer’s requests.

Replies to reviewers

Reviewer #2

Review on the article Effect of different timings of implant insertion on the bone remodeling volume around patients’ maxillary single implants: a 2-3 years follow-up, submitted to International Journal of Environmental Research and Public Health.

The article is well-written and should be considered to be published within the journal. However, during reading, some doubts about certain statements might appear. If possible, the authors should consider making slight changes within the manuscript or provide answers in a response letter. Nevertheless, these doubts do not reduce the overall, positive impression of the article.

  1. Abstract

I would suggest describing healing time period in the case of delayed implant placement.

Our response: According to reviewer's suggestion, the abstract has been modified as requested.

  1. Materials and Methods

Font type in the last sentence of 2.1 section is different in comparison to the rest of the manuscript.

Our response: Thank you for suggestion, but we think the typesetting will be checked by the editorial staff.

Was there any age of the patient that was excluding patient from the research? As presented in some studies, age can negatively influence healing possibilities and overall bone quality.

Our response: Certainly the age of patients was a factor that certainly affects the outcomes. However a multivariate analysis on all the competing factors would require a higher number of participants.

Was Volume of Interest subjected to any smoothing procedures? Initially created anatomical models are characterised most of the time by the presence of several discontinuities as well as unexpected curvatures.

Our response: there isn’t any smoothing adjustment. Remember that this is a single implant site examination. So, the cross-sectional curve from which the Volume Of Interest is obtained is a perfect straight line; and subsequently, the VOI is a regular box in which volume was measured with the maximum definition as possible (depending on the dimension of the voxel).

According to “Implant is placed 1-2 months after tooth extraction” – a month of difference in delayed placement method seems to be significant. Again, some studies present that bone density can change in a duration of a month by a few percent. Did authors observe differences in bone quality between patients subjected to surgeries after 1 and  2 month respectively?

Our response: unfortunately our measurements were collected pre-operatively and at a 1-/3-year of survey. We have no information regarding the status of bone density at 1 or 2 months. Moreover the procedure of measurement with the dentascan at our disposal (Simplant) allowed to measure the volume and density with two independent procedures.

In the last sentence of anatomical variables section, again there is a different font type in comparison to the manuscript style.

Our response: Thank you for suggestion, but we think the typesetting will be checked by the editorial staff.

There is no real description of the implants used in the research. Can authors describe their types (press-fit/threaded) as well as manufacturing procedures with special focus on surface condition? Appropriate coating, surface roughness, material etc. should be described as these factors directly affect the possibility of achieving proper osseointegration.

Our response: According to reviewer's suggestion, the brand of dental implant (with features) was inserted into materials and methods subsection. The uniformity of the implant brand and merchandizing could not adversely affect the variability of the results, so it was one of the strong points of the present paper.

  1. Results

The authors present patients mean age of 46.1 years. What was the age of the youngest and the oldest participant? Their age would significantly affect a mean result. I would suggest presenting, despite of mean age, the number of participants in intervals (i.e. 18-30 years: 3 patients, 30-40 years: 5 patients, etc.). This would give appropriate insight into the age of participants.

Our response: According to reviewer's suggestion, the age range of the participants has been inserted as requested.

  1. Discussion

Successful osseointegration does not depend only on appropriate amount of bone around the implants. There are more factors that affect this phenomena, especially associated with implant construction. For this reason, I would suggest omitting information presented in the fourth paragraph.

Our response: According to reviewer's suggestion, the sentence in the fourth paragraph has been deleted.

Reviewer 3 Report

Very interesting topic, well designed and well written paper.

1. but discussion should be implemented underlying as immediate placement and restoration of a single implant can be a valid and successful option of treatment in the case of single compromised teeth.

2. Moreover, this treatment protocol eliminates the need for removable provisional restoration and seems to maintain the preexisting architecture of soft and hard tissues in most cases.

3. In the discussion Barone A et al j Periodontology 2006 should be cited.

4. Then CBCT should be cited in the discussion comparing results of the study with other studies using periapical xrays ( Quaranta A et al Implant dentistry 2016) and CBCT ( Bruschi E et al. Int J Periodontics Restorative Dent. 2019).

Author Response

Dear Sirs

First of all, I would like to express my appreciation for the comments and suggestions offered by Editor and 4 Reviewers.

We have tried to address all the concerns expressed. To assist you in your work, we have highlighted all changes to the manuscript using the MS Word tracking function, though the modifications are so extensive as to make following the highlighted document all but impossible.

The manuscript has been revised according to the reviewer’s requests.

Replies to reviewers

Reviewer #3

Very interesting topic, well designed and well written paper.

  1. but discussion should be implemented underlying as immediate placement and restoration of a single implant can be a valid and successful option of treatment in the case of single compromised teeth.

Our response: According to reviewer's suggestion, the discussion has been implemented as requested.

  1. Moreover, this treatment protocol eliminates the need for removable provisional restoration and seems to maintain the preexisting architecture of soft and hard tissues in most cases.

Our response: According to reviewer's suggestion, the discussion has been modified as requested.

  1. In the discussion Barone A et al j Periodontology 2006 should be cited.

Our response: According to reviewer's suggestion, the abovementioned reference has been inserted in the discussion subsection.

  1. Then CBCT should be cited in the discussion comparing results of the study with other studies using periapical xrays ( Quaranta A et al Implant dentistry 2016) and CBCT ( Bruschi E et al. Int J Periodontics Restorative Dent. 2019).

Our response: According to reviewer's suggestion, the abovementioned references has been inserted in the discussion subsection, and our results were compared to them

Reviewer 4 Report

Well written, no further comments than to add size legends in Figure 1

Author Response

Dear Sirs

First of all, I would like to express my appreciation for the comments and suggestions offered by Editor and 4 Reviewers.

We have tried to address all the concerns expressed. To assist you in your work, we have highlighted all changes to the manuscript using the MS Word tracking function, though the modifications are so extensive as to make following the highlighted document all but impossible.

The manuscript has been revised according to the reviewer’s requests.

Replies to reviewers

Reviewer #4

Well written, no further comments than to add size legends in Figure 1

Reply to reviewer: dimension of the thiks in each cross-sectional image has been inserted into the figure caption.

Round 2

Reviewer 1 Report

Thank you for giving me this opportunity to re-review your revised manuscript.

I tried to evaluate a carefully modified paper.

Unfortunately, however, Supplementary Materials were not provided in the proper form.

That's why I can't evaluate your manuscript.

Please provide it in the proper form again.

Note: Reviewer 1 gives the decision of "accept" via email after checking the supplementary materials, the comments and decision are as followed: ---------------------   "Thank you for your detailed email reply.   The review comments are listed below.   This paper is worth accepting.   Thank you for giving me this opportunity to re-review your revised manuscript. I am happy that all of the suggested corrections have been made. Thank you for spending so much effort.   Thank you for your consideration."